

# Breaking supercontinents; no need to choose between passive or active

Martin Wolstencroft[1,2] and J. Huw Davies[2]

[1]JBA Risk Management, Skipton, UK
[2]School of Earth and Ocean Sciences, Cardiff University, CF10 3YE, Wales, UK

*Correspondence to*: J. Huw Davies (DaviesJH2@cardiff.ac.uk)

**Abstract.** Much debate has centred on whether continental break-up is predominantly caused by active upwelling in the mantle (e.g. plumes) or by long-range extensional stresses in the lithosphere. We propose the hypothesis that global supercontinent break-up events should always involve both. The fundamental principle involved is the conservation of mass within the spherical shell of the mantle, which requires a return flow for any major upwelling beneath a supercontinent. This shallow horizontal return flow away from the locus of upwelling produces extensional stress. We demonstrate this principle with numerical models, which simultaneously exhibit both upwellings and significant lateral flow in the upper mantle. For non-global break-up the geometry of the mantle will be less influential, weakening this process. This observation should motivate future studies of continental break-up to explicitly consider the global perspective, even when observations or models are of regional extent.

## 1 Introduction

Continental break-up leading to new ocean basins has been a fundamental component of the plate tectonic system since at least the late Proterozoic. The geologic record provides evidence that continents are assembled into larger supercontinents and subsequently broken apart in a cyclical manner (e.g. Wilson, 1966; Bleeker, 2003; Rogers and Santosh, 2003; Bradley, 2011). The source of the force that 'breaks' a continent is of particular interest and continues to be actively studied (Gao et al. 2013; Buiter & Torsvik 2014; Koptev et al. 2015).

This paper does not attempt to explain the whole history and mechanics of plate tectonic history, nor does it consider the fine detail of crustal fracture processes. Instead, we consider the solid Earth system of lithosphere and mantle as a dynamic whole and present the implications of this viewpoint for the large-scale mechanics of continental break-up. Our discussion is mediated by the use of realistic scale and geometry numerical models of mantle circulation.



## 2 Previous Work

Long-range plate-mediated extensional tectonic forces (White and McKenzie 1989) and uplift forces produced by thermally or chemically buoyant mantle (Hooper, 1990; M. Storey, 1995) have both been proposed as candidate mechanisms to drive

continental break-up. In the literature, these two mechanisms developed into end-member hypotheses: a 'passive' model, which relies on extensional stresses and an 'active' mechanism, which involves a thermally buoyant feature underneath a continent. The latter is also known as the 'plume model'. The passive/active terminology originated with Şengör and Burke (1978) and was widely used or implied in the subsequent literature (e.g. Turcotte and Emerman 1983; Bott 1992; 1995; Huismans et al., 2001; Allen and Allen, 2005).


Evidence, which might discriminate between the proposed break-up mechanisms, is equivocal as lithospheric extension and plume head-like activity seems to be related in a complex manner. White and McKenzie (1989) favour extension as the main driver of break-up, proposing that the volcanism associated with continental break-up (e.g. Central Atlantic Magmatic Province) is related to higher mantle temperatures, which develop under large continents through insulation  (e.g. Gurnis,

1988), although this is not universally accepted (e.g. Heron and Lowman, 2014). Experimental results have demonstrated how lithospheric thinning and dyke-like volcanism could be linked to thermo-chemical instability of the lower lithosphere in moderately old cratons (Fourel et al. 2013). Continental break-up appears to occur preferentially on alignments of previous continental collision, suggesting that strain localizes at lithospheric weak points (Corti et al. 2007); favouring the passive model. However, in a classic modelling study, Bott (1992) concluded that simply 'passive' upwelling of mantle in response

to local lithospheric thinning cannot initiate break-up; a more significant source of stress - such as a plume - is required.

The presence of volcanism has been used to argue that mantle plumes actively cause break-up (M. Storey et al. 1995), although there are examples where plume magmatism has not resulted in break-up (Sobolev et al. 2011). B. C. Storey (1995) concluded that the break-up of Gondwana proceeded both with voluminous volcanism and without. Ziegler and Cloetingh

(2004) report large variations in the duration of break-up, from effectively instant break-up to many 10's of Myr of rifting prior to break-up. Cloetingh et al. (2013) suggest that plumes modify lithosphere strength and help initiate break-up in an extensional setting, producing the classic 'plume head' effects. This evidence of combined extensional and plume activity argues for an active plus passive mode, with the simultaneous occurrence of a hot upwelling feature and continental-scale lithospheric extension. Bott (1992; 1995) speculated that such a combined process may be required to achieve full break-up,

but how this might occur was left unresolved.

The assumption that drag forces exerted by mantle flow on the lithosphere are too small to influence plate motions (Forsyth and Uyeda, 1975) influenced subsequent debate. This assumption has been questioned by active-source seismic tomography studies (Kodaira et al. 2014), which indicate that mantle flow may influence plate motion. Sophisticated numerical



modelling studies also suggest that large-scale mantle flow may act as a 'conveyor belt', with plate motions influenced by flow away from active upwelling (Becker and Faccena, 2011; Cande and Stegman, 2011). Therefore, it seems possible that large-scale lateral flow in the upper mantle is capable of producing stress in the lithosphere in certain scenarios. It is interesting to consider whether certain dynamic mantle behaviours could set up such large scale flow and thereby set up the necessary physical conditions for continental break-up to occur.


Long-timescale mantle convection is difficult to constrain empirically and extensive use has been made of numerical modelling. Previous studies of mantle convection in both 2-dimensional and spherical geometry have shown that Earth's mantle is or has been transitionally layered about the 660 km deep Olivine phase boundary (Davies 1995; Yanagisawa et al. 2010; Wolstencroft and Davies 2011; Herein et al. 2013). The transitionally layered state demonstrates time dependent

behavior such as mantle avalanches (Tackley et al. 1993). Although the sources of time dependent behavior in the real Earth may be different, in this paper, avalanches are used as an example of a global-scale kinematic event in the mantle.

## 3 Modelling Method

Modelling was carried out using the TERRA spherical geometry mantle convection model (Baumgardner, 1985; Bunge et al., 1997; Oldham and Davies 2004; Davies and Davies, 2009; Wolstencroft and Davies, 2011). The values given in Table 1

were held constant between model cases, while Table 2 contains parameters which were varied.

Cases were permitted to evolve to a state where there was no long-term trend in heat flux through the mantle. Mantle rheology comprised: uniform, radially variable (after: Bunge et al., 1997) and temperature dependent viscosity. The phase transition in the olivine system from Ringwoodite to Bridgmanite and Ferropericlase at 660 km ('660') was modelled using the sheet mass anomaly approach (Tackley et al. 1993), where buoyancy forces are applied to approximate the resistive

effect of the 660 phase change on mantle flow (Christensen and Yuen, 1985).  The vigour of convection is expressed as the basally heated Rayleigh Number (Ra).

$$Ra = (\alpha\rho g\Delta TD^{3})/\kappa\eta \qquad (1)$$

where: $\alpha$ is the coefficient of thermal expansion, $\rho$ is average density of the two phases, g is gravitational acceleration, $\Delta T$ is the temperature change across the mantle,  D is mantle thickness, $\kappa$ is thermal diffusivity and $\eta$ is dynamic viscosity.

High Ra produces smaller convective features, which have greater relative difficulty breaking through the phase change (Peltier, 1996). When running models at Ra lower than Earth-like ($\sim 10^{8}$, Weeraratne and Manga, 1998) the modelled

Clapeyron slope must be more negative to obtain Earth-like behaviour. The probable value of the real Clapeyron slope for 660 is $-2.5 \pm 0.4$ MPaK$^{-1}$ (Ye et al., 2014).



### 3.1 Depth Dependent Viscosity

Radial viscosity variations were set with a radially varying multiple of the reference viscosity (Table 2, after: Bunge et al. 1997). The transition from upper to lower mantle occurs at 660 km depth. There is a stepped increase into the lower mantle across 660 km, which is consistent with interpretations of Earth's real viscosity profile (Mitrovica and Forte, 2004).

### 3.2 Temperature Dependent Viscosity

For the case with temperature dependence (Case 3), the assumed temperature dependence of viscosity is set by:

$$\eta(\vec{r},T)= \eta(r)\, e^{(4.6(0.5-T))} \qquad (2)$$

where: T is the temperature normalized by the temperature change across the mantle, $\eta(\vec{r})$ the three-dimensional viscosity field and $\eta(r)$ the radial viscosity profile. This relation allows viscosity variation up to a factor of 100, with a lower limit on the viscosity set to 2 orders of magnitude below the reference viscosity to ensure numerical stability of the model. For case 3 the mean viscosity of the whole mantle is used to calculate the Rayleigh Number (Equation 1).

### 4 Modelling Output

We present three example model cases (Table 2). Cases 1 and 3 use a Ra that is 1-2 orders of magnitude lower than Earth and a ~3x more negative 660 Clapeyron slope; case 2 uses a near-Earth-like Ra and a 660 Clapeyron slope slightly more negative than Earth. Using both scaled-down and near Earth-like vigor gives us greater confidence that the dynamic processes modelled are plausible. Figure 1 demonstrates one defining feature of the transitional convective regime; periodic spikes in surface heat flux.

Figure 1 here

Figure 2 presents a detailed visualization of the first spike of Case 1. The event causing the spikes in surface heat flux proceed as follows: cold material that has ponded in the upper mantle overcomes the resistance of 660 and avalanches into the lower mantle. The avalanche partially overturns the whole mantle, advecting a 'pulse' of hot material into the upper mantle. This hot material cools rapidly - the surface heat flux spike. In terms of motion, this event produces globally organized inward flow at the surface towards the avalanche and outward flow above the antipodal upwelling. This motion is demonstrated in Supplementary Animation 1.

Figure 2 Here



Figure 3 demonstrates the same process at high Ra in case 2 and demonstrates the universality of the global pattern. The global nature of these events is required by the inescapable conservation of mass; avalanches must have a return flow. Being rooted in fundamental physics of the finite mantle system this will apply universally, not just under the conditions of these illustrative cases.

Figure 3 here

## 5 Discussion

The debate between 'passive' and 'active' models of continental break-up probably represents a false dichotomy, as continental break-up seems to display or indeed require characteristics of both mechanisms (e.g. Bott 1992; B. C. Storey 1995). Our modelling provides an example of how this might occur, through dynamic events, which impact a large proportion of Earth's mantle for a geologically significant time. The most important aspect of our conceptual model is the return flow required by conservation of mass on a global scale. In the subsections below we consider several aspects and implications related to this central idea.

### 5.1 Force Estimates

The viscous lithosphere used in this study cannot address the details of rifting, accordingly we only require the standard extension processes invoked in more detailed models (Huismans et al., 2001), i.e. far-field plate forces and upwelling generated forces to effect continental break-up. A first order estimate of the driving force for extension arising from an active uplift is $<4\times10^{12}$ Nm$^{-1}$, a similar estimate of the driving force for extension from passive distant plate forces is $<3\times10^{12}$ Nm$^{-1}$ (subduction suction) or $<5\times10^{12}$ Nm$^{-1}$ (subduction slab pull) (Kusznir, 1991). Estimates of the strength of lithosphere are sensitive to its temperature and crustal thickness and range from $2 - 20 \times10^{12}$ N m$^{-1}$ using simple strength envelope assumptions with constant velocity (Davies and von Blanckenburg, 1998) and $4 - 9 \times10^{12}$ N m$^{-1}$ for low strain rates (Stamps et al., 2010).

Comparing the values above, it is unsurprising that break-up might require both processes, since their combined forces ($<8 \times10^{12}$ N m$^{-1}$) are more likely to exceed the integrated strength (average range: $6.5 - 9 \times 10^{12}$ N m$^{-1}$), as argued by Bott (1992). Going beyond such simple estimates requires complex rheology (Burov, 2011), deformation history and damage (Bercovici and Ricard, 2014) and combined plume and far-field stresses in ultra-high resolution (Burov and Gerya, 2014), details that go beyond the scope of this discussion.





### 5.2 Magmatism

During break-up under the model presented here, margin segments located near active upwellings would show evidence of extensive magmatism with a deeper mantle signature as suggested by the plume model. Margin segments along-strike, where upwelling is not as concentrated, would be dominated by extension and show little or no pre-break-up magmatic signature.

Thus observations of both volcanic and non-volcanic margins during break-up can be satisfied (e.g. B. C. Storey 1995).

### 5.3 Time scales

Our modelling is consistent with previous studies of time-dependent or cyclical behaviour of the mantle (Sutton 1963; Davies 1995) and many numerical models have demonstrated mantle avalanches (Machetel and Weber, 1991; Tackley et al. 1993; O'Neill et al., 2007; Herein et al. 2013). We have demonstrated that the behaviour is evident across a range of Ra for

an isoviscous rheology and for a case with depth and temperature-dependent viscosity. Further rheological variations are possible but are beyond the scope of this study. For example, Höink et al. (2012) describe how interactions between a high viscosity lithosphere and a low viscosity asthenosphere can lead to lithosphere stress amplification; a process, which could enhance the ability of mantle convection to promote break-up.

Analysis of the cycle of avalanche behaviour highlights the importance of the return flow (the pulse) to the surface (e.g. Condie, 1998). The avalanche-pulse mechanism has the potential to produce lithospheric stress through both large horizontal near-surface motion and temporally associated plumes. From the models presented, we estimate that the duration of such an event is of order 10's of Myr, a timescale comparable to the break-up of a continent and the opening of a new ocean.

### 5.4 Break up mechanics

It is clear that continental break-up can only be achieved if there is localization of deformation (Le Pourhiet et al., 2013; Moresi et al., 2007; Regenauer-Lieb et al., 2008). This is achieved by feedback and possibly the presence of magma (Corti et al., 2003). There is strong evidence that deformation localizes frequently on regions that have an inherited weakness since they were the sites of earlier continental break-up (Audet and Bürgmann, 2011; Buiter and Torsvik, 2014). Equally while the simple estimates above considered the driving force that an upwelling can provide, hot upwellings can also lead to

magmatism and this can help to weaken the lithosphere including by diking (Bialas et al., 2010; Brune et al., 2013), so the upwelling might not just be important for rifting by providing extra driving force for extension but it might also critically weaken the continental lithosphere. We note that full spherical models incorporating more detailed lithospheric rheologies are only just starting to appear (van Heck and Tackley, 2008, Rolf et al., 2012), and it is only through further advances in such models that a more quantitative assessment of this hypothesis will be achieved.





The fundamental kinematics of the global situation that we demonstrate can also be seen in the modelling of Zhong et al. (2007), who associated supercontinent cycles with low spherical harmonic degree convective structure. However, our specific degree-1 scenario (Figure 4A) appears to be different to the degree-2 break-up scenario proposed by Zhong et al. (2007) (Figure 4B). It is likely that this difference is a matter of interpretation, since plate-driven extension is generated by slab suction from fringing subduction, as well as from distant plate motions, plate driven extension represents a common component.

Figure 4 here

## 6 Summary

Modelling has many limitations; we do not claim that the avalanche-pulse mechanism discussed above is essential for continental break-up; episodic tectonics could have a range of sources (e.g. O'Neill et al., 2007). Even without the avalanche mechanism, we speculate that the return flow required by mass conservation as plumes rise may lead to extensional stresses in the lithosphere. In reality we might expect a more complex situation, with multiple length scales of convection existing on differing timescales (e.g. compare Cases 1-3).

The effectiveness of the mechanism may also depend on continent size. The role of global-scale conservation of mass in organizing sub-plate horizontal mantle flow between upwellings and downwellings is significant because of the hemispheric scale of supercontinents. If the continents being considered are smaller, the impact would be weaker and more subject to variability in the location of upwellings and downwellings. For the break-up of the smallest continental fragments, the geometry of the flow conserving mass might not be reinforcing in terms of horizontal stress, on this basis small continents should be more difficult to break.

Considering continental break-up as a global-scale geodynamic event involving both mantle and crust has clear advantages. This dynamic behaviour is capable of:

- Exerting extensional stresses over long timescales over large areas
- Delivering plume-like features from below in a more spatially discontinuous manner

This variable temporal and spatial relationship between extensional stressing and plume arrival could produce the observed variation in apparent rifting mode. As the process described is most applicable to large-scale continent break-up, if a supercontinent is successfully broken-up we would expect both passive and active drivers to be identifiable. Continental break-up modelling is an active field (e.g. Allken et al., 2011; Gueydan and Précigout, 2014); however, integrating regional





lithospheric and global convection models is a significant challenge. We suggest that future modelling studies should strive to include realistic-scale return flows to place continental break-up in the correct global context.

## Acknowledgements

The authors would like to thank: D. Rhodri Davies, Ian T. Jardine, Glenn Milne, Peter Webb, David Oldham, and Peter Bollada. Modelling was carried out on HECToR, the UK National Supercomputer and Merlin at Cardiff University. Part of the work was supported by NERC: NER/S/A/2005/13131.

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



**Tables**

**Table 1. Common input values. *Since the model is incompressible, the adiabatic temperature gradient would need to be added for**
**comparison to Earth core mantle boundary temperature.**

| Parameter | Value |
| --- | --- |
| Equation of state | Incompressible & Boussinesq |
| Reference density | 4500 kg m$^{-3}$ |
| Gravitational acceleration | 10 m s$^{-2}$ |
| Vol. coefficient of thermal expansion | $2.5 \times 10^{-5}$ K$^{-1}$ |
| Thermal conductivity | 4 W m$^{-1}$ K$^{-1}$ |
| Specific heat (constant volume) | 1000 J K$^{-1}$ kg$^{-1}$ |
| Temperature at surface | 300 K |
| Temperature at CMB* | 2850 K |
| Radioactive heating | $5 \times 10^{-12}$ W kg$^{-1}$ |
| Velocity boundary conditions | Free slip |
| Inner radius of shell | $3.480 \times 10^{6}$ m |
| Outer radius of shell | $6.370 \times 10^{6}$ m |





**Table 2. Model cases. Cl660 is the Clapeyron slope at 660; Ra is Rayleigh number (eq. 1). U/LMV is the upper-lower mantle viscosity contrast, TdV indicates temperature dependent viscosity. *Visualized in Figure 2 and Supplementary Animation 1;**
**†Visualized in Figure 3.**

| Case | Cl660 (MPa K⁻¹) | Ra (Base Heated) | Notes |
|------|------|------|------|
| *1 | -8 | $6.76 \times 10^6$ | Isoviscous |
| †2 | -4 | $8.48 \times 10^7$ | Isoviscous |
| 3 | -8 | $6.97 \times 10^6$ | 10x U/LMV, TdV |







Figure 1: Model surface heat flux time series for the 3 cases presented, truncated to exclude variations as the model stabilises the
initial condition. The surface heat flux magnitude variations and timing offsets are the result of the varying vigour of convection.





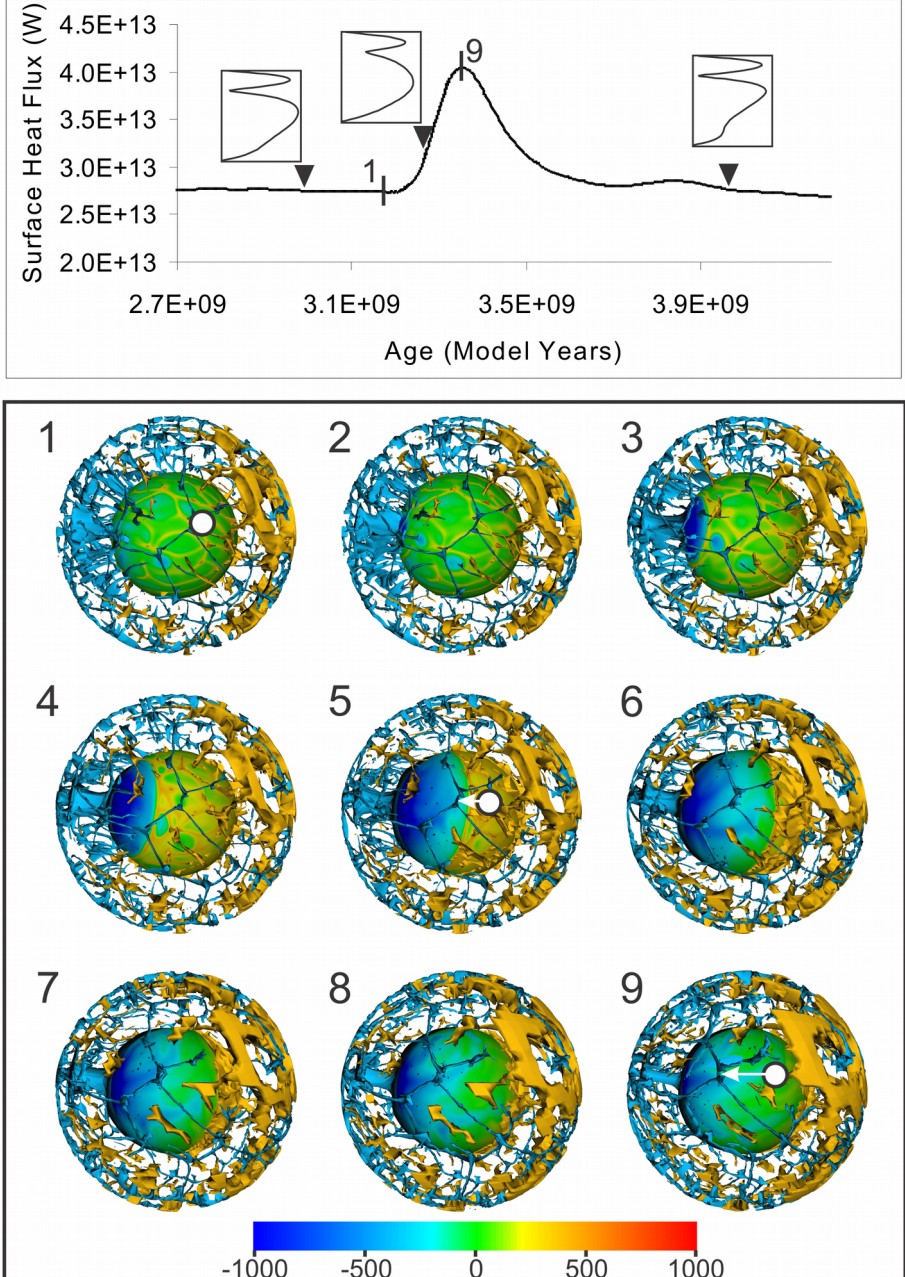

**Figure 2: Visualisation of Case 2, peak 1 shown in Figure 1. In addition to surface heat flux, the graph provides 3 insets showing the absolute radial mass flux through the modelled mantle, the depth of the 'pinch point' indicating how much 660 is restricting mass exchange between upper and lower mantle. Evenly spaced temperature anomaly snapshots (1-9) cover the indicated region of the surface heat flux curve. The white dot in 1, 5 and 9 is fixed and arrows show how the near surface material has moved. Temperature anomaly is plotted just above the CMB and as ±400 K isosurfaces.**



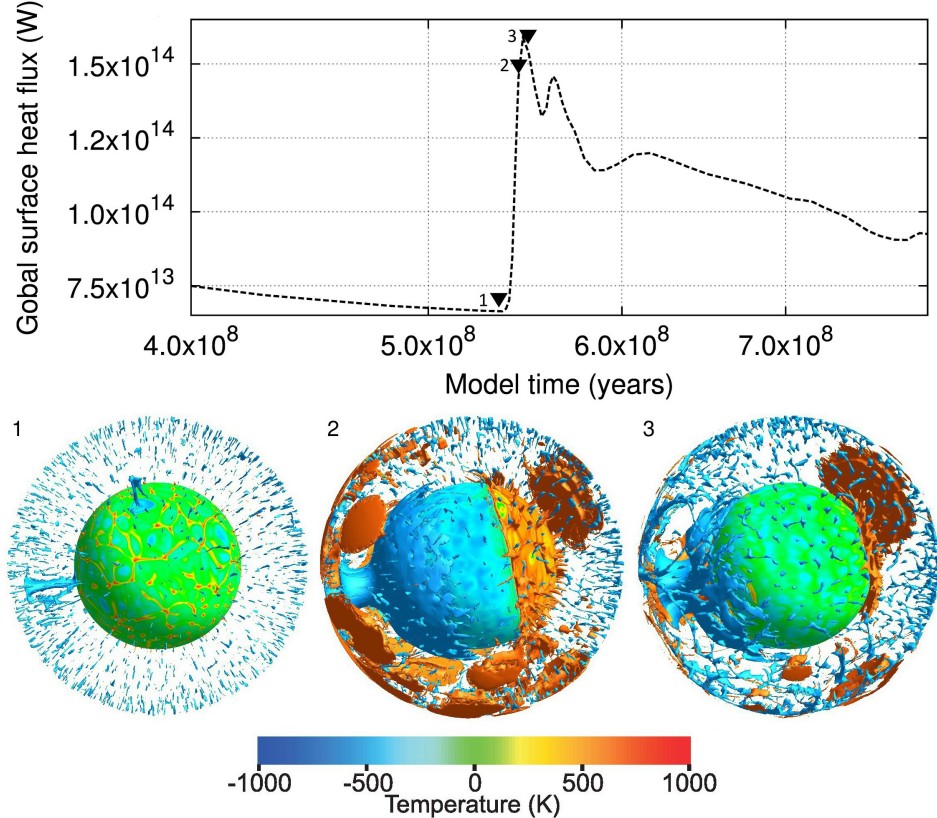

**Figure 3: Visualization of Case 2. The graph (top) is an enlargement from Figure 1, the numbered triangles indicate the time of the panels below. 1) The avalanche initiates, material from the upper mantle begins to enter the lower mantle where it shows up as anomalously cold. 2) Avalanche in full flow, antipodal plumes have already reached the surface. 3) The avalanche has now progressed to 'pulling' hot material from the antipodal plume towards itself. Isosurfaces follow -500 and +700 K anomalies, uppermost 5% of the isosurfaces clipped to improve clarity of the deep mantle.**





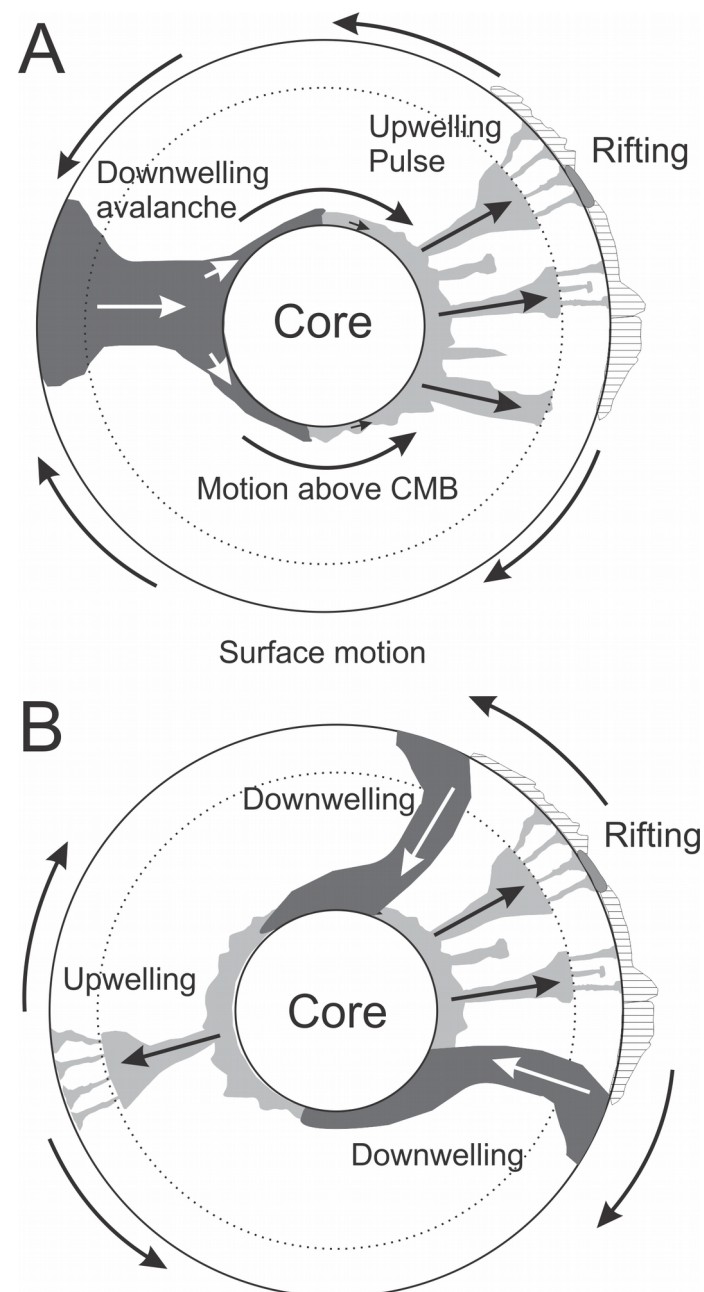

**Figure 4: Conceptual sketch of the proposed mechanism of break-up. (A) Interpretation from Cases 1 and 2. (B) A further possible**
**configuration of large scale return flow involving continent fringing subduction zones (after: Zhong et al., 2007). Not to Scale.**
