# Peer review of "Breaking supercontinents; no need to choose between passive or active"

_Solid Earth, 2017_

## Referee Comment (RC1) · Anonymous Referee #1 · 14 Mar 2017

This paper deals with the topic of (super-)continental break-up in a global – i.e. mantle dynamics – perspective based on spherical mantle convection computations. The more specific focus is the "ambiguity" of passive and active mechanisms in this process and the fundamental importance of sublithospheric return flows induced by dynamic processes, here mantle downwelling avalanches. I think this work generally touches on an important topic in geodynamics, particularly for our understanding of the interplay between mantle convection and surface tectonics. Consequently, Solid Earth is an appropriate journal for publication of this work.The manuscript is generally easy to follow and in most cases the argumentation of the authors is clear. However, I do have some moderate concerns related to the originality of the proposed concepts, the

choice/setup of the numerical simulations and their analysis, and the integration into existing geodynamical concepts. I think these concerns (which I detail below) require moderate revisions before publication of the manuscript.

Specific comments:

1.) The general idea behind this paper (no clear distinction between active and passive break-up mechanisms) seems not really novel. In fact, it is somewhat trivial that any break-up process requires extension in the lithosphere (how else would you do it?). The real point, however, is the cause for the extension. It could be induced by mantle upwellings (e.g. plumes) or alternatively by "far-field" processes, e.g. some more or less remote subduction (see e.g. Bercovici & Long, 2014). I think this has to be clarified throughout the manuscript including the first sentence of the abstract.

2.) Just 3 convection simulations are presented. Ok, such calculations are computationally very expensive, but this is still a little disappointing. Moreover, 2 of the 3 models are isoviscous cases whose geodynamic relevance is very limited as they e.g. fail to generate a strong lithosphere. However, many geodynamic studies have demonstrated that surface strength is very relevant for breaking the lithosphere (if continental or not). To just list a few: Yoshida (2008, GRL, 25, L23302), Rolf et al. (2014, GRL, 41, 2351-2358). Only case 3 thus seems to have good geodynamic relevance, although 2 orders of magnitude viscosity variation is probably not enough to describe an Earth-like convection regime (Solomatov, 1995, Phys. Fluids, 7, 266-274). In addition, when I look at the time series of case 3 in Figure 1, I'm quite unsure if it has reached equilibrium or is still in some sort of transient state. How can you be sure about that? Just based on the heat flow evolution?

Finally, it is unfortunate to see that the analysis jumps directly from isoviscous cases to one with both layered AND temperature-dependent viscosity, rather than including a case with only layered, but NOT temperature-dependent viscosity, just to see the effects of that. Do you expect that the lateral variations induced by temperature-

dependent viscosity (which finally determine the strength of sinking slabs) are important for avalanches to occur or not and thus have implications for your discussion?

3.) The chosen Clapeyron slopes are extremely large. You explain this choice by the reduced convective vigour in the models. For the high Ra case, however, surface heat flow is ∼85 TW, i.e. roughly 2x Earth's, so convective vigour actually seems higher than Earth's. For the other 2 cases, heat flow is lower than Earth's, but not much. My point is that to make avalanches possible in your model you seem to need very large Clapeyron slopes and I don't think the argument of reduced convective vigour can fully explain that (unless the heat flow comparison is not a good proxy for convective vigour here?). As far as I understood, your model does not feature a density jump across the 660. If you include that, may it be easier to pile up cold material on the 660 using smaller Clapeyron slopes? To be clear, I don't ask for additional cases with smaller Clapeyron slopes here, but I encourage the authors to discuss the shortcomings of their model setup and how they may affect their conclusions in somewhat greater detail. This does not only include the Clapeyron slopes, but also the omission of e.g. compositional variation and the rather small lateral variation in viscosity (see above).

4.) Given that only a very small set of cases is concerned, I hoped to see some more in-depth analysis of those cases at least, but the only presented diagnostic is the average surface heat flow evolution. However, the main physical processes used in the argumentations are a) mantle avalanches and b) return flows. For the former, it would be interesting to see a diagnostic such as the radial mass flux through the 660 as already suggested in Figure 2. The return flow may be more difficult to quantify, but perhaps horizontal velocities at some shallow depth range are an option. Considering the time evolution of such additional measures could clarify causalities here and could even give an idea about how much extensional stress may be induced to the lithosphere (shear stresses/tractions at the base of the lithosphere?).

5.) Again, concerning the conservation of mass: I agree that return flows are required in the described geodynamic settings. However, what is perhaps less known is the

wavelength over which they occur. Here, your models indicate the longest possible wavelength (i.e. degree 1). Later on (Figure 4, lines 181-186), you relax this view by comparing to Zhong et al. (2007), however, this deserves more discussion. What may control this wavelength, etc.? Personally, I find your degree-1 concept (e.g. Figure 4A) problematic, because it seems difficult to have a stationary supercontinent antipodal to a persisting complex of surface convergence (which seems required to pile up material on top of the 660). Instead, I would expect the continent to either move to this convergence zone and/or to break-up before an avalanche may occur. In a shorter-wavelength flow like degree-2, however, the supercontinent may be kept stationary, e.g. by surrounding subduction. This may be discussed a little more.

Further minor comments:

- line 12/13: "For non-global …". This sentence is not really clear (at least at this point). What do you mean with "the geometry of the mantle". The flow pattern?

- line 40: You may want to consider the work of Brandl et al. (2013) in your referencing, which discusses evidence for elevated temperatures below Pangea from a data-based approach: Brandl et al., 2013, Nat. Geosci., 6, 391-394.

- line 48/49: "Storey (1995) concluded …" This sentence is unclear to me. Do you mean that parts of the Gondwana break-up occurred with volcanism while other parts did not?

- line 60ff: Here, it seems relevant to add some discussion about the likely role of continents in plate-mantle coupling. After all, thick continents are likely to increase e.g. the magnitude of shear tractions at the base of the lithosphere (e.g. Zhong, 2001, JGR, 106, 703-712, Conrad & Lithgow-Bertelloni, 2006, GRL, 33, L05312). So, the presence of continents may in a way help to induce stresses in the lithosphere that eventually cause their own break-up.

- line 73ff: Regarding the model description: What is the numerical resolution used in

the TERRA models? Also, I think you should state clearly that while you are interested in continental break-up, continents are not explicitly included in your model.

- line 77ff: Is there any density increase associated with the phase transition at 660 km (see my comment 3 above)? It is also worth to note that you ignore all other phase transitions.

- line 80ff: When defining the Rayleigh number, you use "kappa" (thermal diffusivity), but you don't give its value in Table 1. Ok, it is straight-forward to compute it from the other parameters in that table (kappa = k/rho/c_p), but I suggest to either give this relation somewhere or to list kappa in the table explicitly.

- line 109, This sentence describes the spikes in surface heat flow in Figure 1. While these are easy to spot for case 1, case 2 does not feature them clearly nor does case 3, which seems to feature only one peak, but then does not seem to have the same equilibrium state before and after the peak (heat flow is quite different). So, are these heat flow peaks really characteristic for the discussed regime?

- line 152-155: This paragraph is extremely short and quite speculative. I don't think it is worth to call this a separate section. Since my other comments probably require additions to the manuscript, I actually suggest deleting this paragraph. If you want to keep this section, however, it should be extended and more explicitly linked to the discussed models.

- line 167/168: How do you actually estimate from your results that the events take 10s of Myr? Just from the presented heat flow curves? Also, please add a reference here when you mention that this is comparable to Earth's timescales.

- line 178: I suggest to add a reference to Yoshida's works here, too (e.g. Yoshida & Santosh., 2014, Geosci. Frontiers, 5, 77-81).

- Table 1: Consider to include the symbols for the physical properties used in eq. (1) in the table, e.g. alpha for the thermal expansivity. I would also appreciate values for the

reference viscosity, right now it is only given implicitly via Ra.

- Figure 2: I think this figure would benefit from a time series of the "absolute radial mass flux" similar to the one of surface heat flow as already presented. Ideally, this should be done Figure 1, too, because then the reader could get an idea of how a mantle avalanche is linked to the surface heat flow spikes and perhaps get an idea about timescales, which would improve the (very short) discussion in section 5.3.

Some very minor suggestions:

- line 8: "long-range" –> "long-wavelength" (?)

- line 74: "The values" –> "The parameter values".

- line 93: "set with" –> "set up with" (?)

- line 101: "by the temperature change" –> "by the superadiabatic temperature change"

- line 107: delete "slightly"

- line 175: Check sentence structure: "including by diking"?

---

## Short Comment (SC1) · 15 Mar 2017

The research paper presents novel and interesting results and corresponds to the scope of the Solid Earth. I consider this review very interesting and timely study in which the authors use numerical simulations to propose that supercontinent breakup events should always involve both active mantle upwelling and extensional stresses in the lithosphere. Overall, the paper is quite short, but very well-written, pretty balanced, and the illustrations are to the point. It provides a modern perspective on the topic.

I have some comments to the article:

1) An important aspect that should be mentioned is the mantle structure. In particular,

should be mentioned that the temporal evolution of the convective wavelength is still an enigmatic question, and it has important implications for the location of major mantle upwellings and the initiation of continental breakup. The present-day Earth's mantle structure is dominantly at spherical harmonic degree-2. However, how mantle structure may have evolved in the geological past is still unclear (Zhong et al., 2007). Also, phase transformations in the mid-mantle–which are often ignored in mantle convection simulations–introduce buoyancy forces and thermomechanical effects in the convective mantle system that can fundamentally affect the patterns of mantle convection (Faccenda & Dal Zilio, 2017).

2) I am intrigued by the discussion about the source of these far-field extensional stresses. The authors should consider that subduction and subsequent roll-back of oceanic plates at continental margins have been invoked to support the occurrence of large-scale, long-term extensional stresses (Bercovici & Long, 2014; Lowman & Jarvis, 1996). Also, numerical simulations indicate that downgoing slabs play a central role in establishing the location and formation of subcontinental mantle plumes (Tan et al., 2002; Heron & Lowman, 2011; Heron et al., 2015; Lenardic et al., 2011; Zhang et al., 2010; Zhong et al., 2007).

3) During the last years, the role of deep slab dynamics appears central to triggering breakup of continents. Penetration of the slab into the lower mantle may generate a surge of compression at the plate boundary because (i) trench migration slows down (Goes et al., 2008; Faccenna et al., 2017), and (ii) because the upper plate is dragged against the subduction zone by a large-scale return flow. This return flow seems to be the key ingredient to triggering supercontinent breakup: subduction of lithosphere in the lower mantle reorganises the mantle flow into a wide cell, thereby localising extensional stresses at greater distances from the trench (Dal Zilio et al., 2017), which eventually may culminate in the breakup of supercontinents.

I hope my comments contribute the authors to improve the manuscript.

Best regards, Luca Dal Zilio

References 1) Bercovici, David, & Long, Maureen D. 2014. Slab rollback instability and supercontinent dispersal. Geophysical Research Letters, 41(19), 6659–6666. 2) Dal Zilio, Luca, Faccenda, Manuele, Capitanio, Fabio A. 2017. The role of deep subduction in supercontinent breakup. Tectonophysics. 3) Faccenda, Manuele, & Dal Zilio, Luca. 2017. The role of solid–solid phase transitions in mantle convection. Lithos 268: 198-224. 4) Faccenna, C., Oncken, O., Holt, A. F., & Becker, T. W. 2017. Initiation of the Andean orogeny by lower mantle subduction. Earth and Planetary Science Letters, 463, 189-201. 5) Goes, Saskia, Fabio A. Capitanio, and Gabriele Morra. 2008. Evidence of lower-mantle slab penetration phases in plate motions. Nature 451.7181: 981-984. 6) Heron, Philip J, & Lowman, Julian P. 2011. The effects of supercontinent size and thermal insulation on the formation of mantle plumes. Tectonophysics, 510(1), 28–38. 7) Heron, Philip J, Lowman, Julian P, & Stein, Claudia. 2015. Influences on the positioning of mantle plumes following supercontinent formation. Journal of Geophysical Research: Solid Earth, 120(5), 3628–3648. 8) Lenardic, Adrian, Moresi, L, Jellinek, AM, O'neill, CJ, Cooper, CM, & Lee, CT. 2011. Continents, supercontinents, mantle thermal mixing, and mantle thermal isolation: Theory, numerical simulations, and laboratory experiments. Geochemistry, Geophysics, Geosystems, 12(10). 9) Lowman, Julian P, & Jarvis, Gary T. 1996. Continental collisions in wide aspect ratio and high Rayleigh number two-dimensional mantle convection models. Journal of Geophysical Re- search: Solid Earth, 101(B11), 25485–25497. 10) Tan, Eh, Michael Gurnis, and Lijie Han. Slabs in the lower mantle and their modulation of plume formation. Geochemistry, Geophysics, Geosystems 3.11 (2002): 1-24. 11) Zhang, Nan, Zhong, Shijie, Leng, Wei, & Li, Zheng-Xiang. 2010. A model for the evolution of the Earth's mantle structure since the Early Paleozoic. Journal of Geophysical Research: Solid Earth, 115(B6). 12) Zhong, Shijie, Zhang, Nan, Li, Zheng-Xiang, & Roberts, James H. 2007. Supercontinent cycles, true polar wander, and very long-wavelength mantle convection. Earth and Planetary Science Letters, 261(3), 551–564.

---

## Referee Comment (RC2) · Anonymous Referee #2 · 13 Apr 2017

This paper presents three global mantle convection calculations using a full 3D spherical shell model. The study aims to provide a determination of the cause of supercontinent rifting, arguing that rifting is best explained when both mantle plumes and extensional rifting are present, rather than being attributed to one or the other of these phenomenon. In the context of this study, extensional rifting appears to be adopted as a term to describe rifting that occurs due to a cumulative effect of the tractions associated with convection (e.g., due to the pull associated with distant subduction) on lithospheric stresses (however, this needs clarification). For example, Figure 4b shows downwelling on the margins of a supercontinent, with the suggestion that tractions due to flow towards the subduction zones, below the continent, contribute to intracontinental

'long-range extensional stresses' that cause rifting. The ideas presented are not particularly novel, indeed supercontinent breakup studies have been presented for some 30 years now, including in other much more systematic and comprehensive studies (e.g., Rolf et al., 2014) that also utilized a spherical shell geometry. The newer contribution from this study is the analysis of a mantle avalanche that occurs with the modelling of an endothermic phase change at a depth of 660 km and an argument for the role of avalanches that would contribute to continental rifting to help drive a global supercontinent cycle. This idea is interesting but unfortunately various aspects of the (just) three models presented mean that they are not suitable for modelling the supercontinent cycle or even just continental breakup.

The most fundamental manifestation of mantle convection is the motion of the Earth's tectonic plates. The plates, continent cratons and deep convection comprise a coupled system in which feedback makes it impossible to separate the study of one without the influence of the other. In particular, a global model of terrestrial mantle convection must include plates in order to emulate terrestrial evolution. Moreover, it must include continental cratons in order to model the supercontinent cycle or infer the reasons for its occurrence. The models presented here do not appear to feature plates not incorporate cratons. Indeed, two of the three models presented are isoviscous and the third includes a thermal viscosity contrast of just two orders of magnitude. These parameters do not allow for the generation of plates. Consequently we have to ask what can we learn from models that do not include plates when they are applied to trying to understand a process that affects continental lithosphere (continental rifting).

Although the suggestion that intermittent mantle avalanches may play a role in driving a supercontinent cycle (that necessitates periodic rifting) is worth exploring, in this manuscript the calculations presented for the purpose of supporting this hypothesis are inadequate (in number and sophistication) to adequately address the issue. At a minimum the authors need to demonstrate the surface characteristics of their cases - do they exhibit anything like plate tectonics? The models need to include continental

cratons if they are to analyze the effect of convection on cratons (e.g., if cratons were present would there be a connection between their positions and avalanches). Has there ever been a study that found avalanches occur in global models (2D or 3D) that feature plate-like surface motion and evolution?

The bottom line is that plate tectonics is an observed phenomenon but mantle avalanches are a phenomenon that appears in some mantle convection simulations and no concrete evidence for avalanches in the Earth's history exists. A study claiming that avalanches can play a role in supercontinent breakup should at least include model supercontinents and mantle convection observables (plates) as a starting point.

Further comments on presentation:

1. The figures should include some model geotherms (i.e., laterally averaged temperature as a function of depth).

2. A figure of the surface viscosity field of model 3 should be included.

3. Present core heat flow as well as surface (Figure 1 is heat flow, not flux).

4. What is a 'smaller' convective feature (line 88) and why would it have more trouble breaking through the phase change? Specifically, it should be more vigorous in a high Rayleigh number flow.

5. On line 67 a comparison appears to be made between 2D and spherical shells. Is that 2D Cartesian geometry or 2D annuli geometry (please be clear).

6. On line 118 what exactly is meant by inward and outward as they are used here. This doesn't appear to be referring to radial directions but rather lateral. Can more careful wording by offered?

7. There are many previous studies on supercontinent breakup and identification of its causes that have not received adequate referencing. For example, please check and include some of the papers from the 90s, in particular those that discussed passive

versus active mechanisms for supercontinent rifting.

---

## Author Comment (AC1) · 29 May 2017

Preamble to response:

We are grateful to both anonymous reviewers and the additional review of Dal Zilio for all their constructive comments and have made our best efforts to incorporate all their suggestions where sensible. There is a common theme in the anonymous reviewers comments, which gives this manuscript a goal which it actually does not have. There is no attempt in this paper to claim that mantle avalanches are the cause of super-continent breakup, or to model the breakup process. As described in the abstract, the point is much simpler; that if one has a supercontinent, and a major upwelling beneath, then conservation of mass requires a large-scale horizontal flow away from that region

which will also drive long-range extensional stresses, i.e. far-field or long range, passive processes. Thus we can expect both active (upwelling) and passive (long-range extensional plate forces) processes to play a role. Mantle avalanches are just one way of generating such upwellings and they allow us to illustrate the associated horizontal diverging flow. Therefore, in our response to the reviewers comments we have not accepted their challenge of writing a whole new manuscript that explains global supercontinent break-up and modelling it in detail, but restricted ourselves to the comments that are pertinent to the message of the manuscript. We now address each one of the comments in turn, in detail.

RC1

1.) The general idea behind this paper (no clear distinction between active and passive break-up mechanisms) seems not really novel. In fact, it is somewhat trivial that any break-up process requires extension in the lithosphere (how else would you do it?). The real point, however, is the cause for the extension. It could be induced by mantle upwellings (e.g. plumes) or alternatively by "far-field" processes, e.g. some more or less remote subduction (see e.g. Bercovici & Long, 2014). I think this has to be clarified throughout the manuscript including the first sentence of the abstract.

Response: We disagree that the general idea is not really novel. We have searched and not found a reference that explicitly focusses on this and neither does the reviewer cite such a reference. Therefore we are happy that this idea is both novel and exciting. Break-up obviously requires extension, but that is not our point. We state that this extension can result from long-range extensional stresses – the aforementioned far-field – passive process and active upwelling. So while we agree with the reviewer the real point is the cause of extension, our point is that it is not "could be induced by mantle upwellings or alternatively by "far-field" processes, but that in fact at supercontinent scale, it will frequently be and. We think that "long-range extensional stresses" is a clearer statement of "far-field processes", which can be ambiguous.

2.) Just 3 convection simulations are presented. Ok, such calculations are computationally very expensive, but this is still a little disappointing. Moreover, 2 of the 3 models are isoviscous cases whose geodynamic relevance is very limited as they e.g. fail to generate a strong lithosphere. However, many geodynamic studies have demonstrated that surface strength is very relevant for breaking the lithosphere (if continental or not). To just list a few: Yoshida (2008, GRL, 25, L23302), Rolf et al. (2014, GRL, 41, 2351-2358). Only case 3 thus seems to have good geodynamic relevance, although 2 orders of magnitude viscosity variation is probably not enough to describe an Earth-like convection regime (Solomatov, 1995, Phys. Fluids, 7, 266-274). In addition, when I look at the time series of case 3 in Figure 1, I'm quite unsure if it has reached equilibrium or is still in some sort of transient state. How can you be sure about that? Just based on the heat flow evolution? Finally, it is unfortunate to see that the analysis jumps directly from isoviscous cases to one with both layered AND temperature-dependent viscosity, rather than including a case with only layered, but NOT temperature-dependent viscosity, just to see the effects of that. Do you expect that the lateral variations induced by temperature-dependent viscosity (which finally determine the strength of sinking slabs) are important for avalanches to occur or not and thus have implications for your discussion?

Response: The reviewer is correct, we do not explicitly model the break-up of a supercontinent. We never claim to do that. Rather the objective of the models was to illustrate that upwellings on a global scale will be accompanied by surface lateral flow (far-field process – long-range extensional stresses). The small number of cases suffice to achieve this i.e. to show that active and passive will be combined. Therefore, these models are trying to capture just the largest mantle scale flows – and lower viscosity variations will capture the essence of the sub-lithosphere flows (since as the reviewer points out – the largest viscosity variations are found in the surface plates and boundaries – which these models do not attempt to capture). We will add the references which do attempt to do this (see below)

The Earth has always been secularly evolving, therefore there is no need for relevant simulations to necessarily be at exact thermal equilibrium; and sometimes after perturbations they can take some time to return. What case 3 shows is an upwelling, and then associated surface horizontal flow – again the point of the paper.

We and others have run simulations with just layered viscosity, and what that leads to is larger length scale flow which will strengthen the case made in this manuscript. Since the point of this manuscript was relatively simple – to present a novel, simple (but powerful) idea and illustrate with a few simple simulations; and not model supercontinent break-up the suggestions of further simulations are beyond the bounds of this manuscript. Please also see the text added to the manuscript in response to point 3.

Added to the manuscript: [Section 2] Near surface factors such as plate rheology have also been shown to influence the preferred large scale pattern of convection (e.g. Yoshida, 2008; Rolf el al., 2014). This study does not attempt to simulate this; we do not impose surface or near-surface conditions to simulate plates.

3.) The chosen Clapeyron slopes are extremely large. You explain this choice by the reduced convective vigour in the models. For the high Ra case, however, surface heat flow is #85 TW, i.e. roughly 2x Earth's, so convective vigour actually seems higher than Earth's. For the other 2 cases, heat flow is lower than Earth's, but not much. My point is that to make avalanches possible in your model you seem to need very large Clapeyron slopes and I don't think the argument of reduced convective vigour can fully explain that (unless the heat flow comparison is not a good proxy for convective vigour here?). As far as I understood, your model does not feature a density jump across the 660. If you include that, may it be easier to pile up cold material on the 660 using smaller Clapeyron slopes? To be clear, I don't ask for additional cases with smaller Clapeyron slopes here, but I encourage the authors to discuss the shortcomings of their model setup and how they may affect their conclusions in somewhat greater detail. This does not only include the Clapeyron slopes, but also the omission of e.g. compositional variation and the rather small lateral variation in viscosity (see above).

Response: These are the Clapeyron slopes required for avalanches in our models, but it does not matter whether they are large or small as we do not try to say that this is the process to generate supercontinent break-up. As we state explicitly in the abstract – we just demonstrate the principle that a return flow is required with any major global upwelling.

Heat flow is not a perfect proxy in these models as there is no surface lithosphere – discussed above.

There is no compositional density jump (we believe that the mantle is not layered at this boundary), but there is the phase change density jump and this is included. We have expanded the text to make this clear and added the density change relation.

We have included additional discussion of the limitations of the model in the manuscript. But again we neither try to make the point that these models simulate the whole process and we don't claim that avalanches are the only way to achieved this.

Added to the manuscript: [Section 3] The density increase associated with the phase change at 660 is included (Table 1) and the phase buoyancy parameters (Wolstencroft and Davies 2011; Equation 6) are provided in Table 2.

Value added to Table 1: Density change at 660 – 9.1%

Added to Table 2: Phase buoyancy parameters for each case

[Section 5 – Modelling Limitations] As the models presented are presented as an illustrative selection, there are aspects of the solid Earth system which they either do not capture or capture in a simplified manner. In terms of direct geodynamic relevance, given that all models are inaccurate in some way, we chose to limit our models to avoid the danger of over-interpretation that can occur where models are considered more 'real' e.g. incorporating complex chemical heterogeneity. This leads to some oddities; for example, Case 2 demonstrates a rather high surface heat flux (Figure 3), but this

is a natural consequence of not imposing a high viscosity – or even rigid – simulated lithosphere (Section 2). Thus the absolute value of surface heat flux from our models is not comparable to the Earth. The lack of a simulated lithosphere also means we do not draw conclusions on detailed break-up mechanics in the lithosphere. Our most complex model – Case 3 – implements a layered radial viscosity profile and allows temperature dependent viscosity. These viscosity features were introduced together, as they are somewhat complimentary in the transitionally layered state – a temporarily isolated lower mantle heats up and becomes less viscous.

4.) Given that only a very small set of cases is concerned, I hoped to see some more in-depth analysis of those cases at least, but the only presented diagnostic is the average surface heat flow evolution. However, the main physical processes used in the argumentations are a) mantle avalanches and b) return flows. For the former, it would be interesting to see a diagnostic such as the radial mass flux through the 660 as already suggested in Figure 2. The return flow may be more difficult to quantify, but perhaps horizontal velocities at some shallow depth range are an option. Considering the time evolution of such additional measures could clarify causalities here and could even give an idea about how much extensional stress may be induced to the lithosphere (shear stresses/tractions at the base of the lithosphere?).

Response: This is an interesting suggestion to improve the understanding of the simulations presented; however, we did not intend to imbue these cases with a status of explaining the mechanics of continental breakup. The goal we have for them in the manuscript is just as plausible models that illustrate the principle – which comes across well in the animation (supplementary material). With that caveat we appreciate that readers may want a little more analysis and have added a figure and some accompanying text, paying attention to the horizontal surface velocity. More complete examinations of radial mass flux for such simulations are presented for example in a range of mantle avalanche papers, including our own (Wolstencroft and Davies, SE, 2011), and readers interested in that aspect can find it in this and other papers in the

literature.

Added to the Manuscript: We have introduced a new Figure 4 (attached), caption: Plot of global surface heat flux and horizontal velocity for Case 2. Both curves are plotted with data points at the same time intervals, grey dotted line demonstrates the timing offset between velocity and heat flux increase.

[Section 3 ] Figure 4 demonstrates clearly how the surface velocity of the model changes in response to the avalanche-return flow in case 2. The velocity increase occurs slightly before the increase in surface heat flux, in accordance with the avalanche-then-plume sequence shown for this model (Figure 3).

5.) Again, concerning the conservation of mass: I agree that return flows are required in the described geodynamic settings. However, what is perhaps less known is the wavelength over which they occur. Here, your models indicate the longest possible wavelength (i.e. degree 1). Later on (Figure 4, lines 181-186), you relax this view by comparing to Zhong et al. (2007), however, this deserves more discussion. What may control this wavelength, etc.? Personally, I find your degree-1 concept (e.g. Figure 4A) problematic, because it seems difficult to have a stationary supercontinent antipodal to a persisting complex of surface convergence (which seems required to pile up material on top of the 660). Instead, I would expect the continent to either move to this convergence zone and/or to break-up before an avalanche may occur. In a shorterwavelength flow like degree-2, however, the supercontinent may be kept stationary, e.g. by surrounding subduction. This may be discussed a little more.

Response: We agree that the details of how such processes would work in detail are uncertain, and we have accepted the prompt to discuss it a little more around our discussion of Zhong.

Added to the Manuscript: [Section 5 ln ~190] now reads: The fundamental kinematics of the global situation that we demonstrate can also be seen in the modelling of Zhong et al. (2007), who associated supercontinent cycles with low spherical harmonic de-

gree convective structure. However, our schematic of the specific degree-1 scenario (Figure 5A) appears to be different to the degree-2 break-up scenario proposed by Zhong et al. (2007) (Figure 5B). It is likely that this difference is a matter of interpretation of mantle/lithosphere interaction, since plate-driven extension could be generated by slab suction from fringing subduction, as well as from distant plate motions. Indeed, given that we do not model a supercontinent over our upwelling, it is a reasonable expectation that such a continent would not stay static, but would tend to migrate towards the downwellings even as it breaks up, leading to a more 2-degree mode of convection in the manner of Zhong. The critical point that will hold is that the surface convergence must be away from the surface supercontinent and that the upwelling will impact the supercontinent; plate driven extension represents a common component and our principle of mass conservation would still apply.

Further minor comments: - line 12/13: "For non-global . . .". This sentence is not really clear (at least at this point). What do you mean with "the geometry of the mantle". The flow pattern?

Response: We have clarified this sentence – it is the fact that the globe is a finite sphere is what brings conservation of mass into play for processes (like breakup of 'global supercontinents) which impact a significant portion of the sphere.

Manuscript now reads: For non-global break-up the impact of the finite geometry of the mantle will be less pronounced, weakening this process.

- line 40: You may want to consider the work of Brandl et al. (2013) in your referencing, which discusses evidence for elevated temperatures below Pangea from a data-based approach: Brandl et al., 2013, Nat. Geosci., 6, 391-394.

Response: Thank you for this suggestion, reference added.

- line 48/49: "Storey (1995) concluded . . ." This sentence is unclear to me. Do you mean that parts of the Gondwana break-up occurred with volcanism while other parts

did not?

Response: Yes – this is what we meant, we have altered the text a little to make this clearer

Manuscript now reads: B. C. Storey (1995) concluded that some regions the break-up of Gondwana underwent break-up proceeded both with voluminous volcanism but other regions without.

- line 60ff: Here, it seems relevant to add some discussion about the likely role of continents in plate-mantle coupling. After all, thick continents are likely to increase e.g. the magnitude of shear tractions at the base of the lithosphere (e.g. Zhong, 2001, JGR, 106, 703-712, Conrad & Lithgow-Bertelloni, 2006, GRL, 33, L05312). So, the presence of continents may in a way help to induce stresses in the lithosphere that eventually cause their own break-up.

Response: Yes, if this work was to be extended, it would be along the many lines suggested by this reviewer. But as we point out above, this is beyond the intended scope of this manuscript. Therefore, while we refrain from adding this discussion to the Introduction, we do feel that this point is worthy of an airing in the context of this work. We have therefore added a little relevant text to the discussion.

Added to the Manuscript: [Section 5 $\sim$ ln 190] The greater thickness of continental vs ocean lithosphere may also act to magnify stresses through increased shear tractions between asthenosphere and lithosphere (e.g. Conrad and Lithgow-Bertelloni, 2006 and references therein).

- line 73ff: Regarding the model description: What is the numerical resolution used in the TERRA models? Also, I think you should state clearly that while you are interested in continental break-up, continents are not explicitly included in your model.

Response: We have included this information.

Added to the manuscript: [Section 3] Model resolution was $\sim$22 km for Cases 1 and 2

and ∼44 km for Case 3; continental material was not modelled.

- line 77ff: Is there any density increase associated with the phase transition at 660 km (see my comment 3 above)? It is also worth to note that you ignore all other phase transitions.

Response: Yes, it is associated, as mentioned in comment 3 above we now said explicitly and added the value to Table 1. We have also commented that we ignore all other phase transitions

Added to the manuscript: [Section 3 ∼ln 85] No other phase changes were modelled.

- line 80ff: When defining the Rayleigh number, you use "kappa" (thermal diffusivity), but you don0t give its value in Table 1. Ok, it is straight-forward to compute it from the other parameters in that table (kappa = k/rho/c_p), but I suggest to either give this relation somewhere or to list kappa in the table explicitly.

Response: Included [∼ln 90] ...and $\kappa$ is thermal diffusivity = k/Cp where: k is thermal conductivity and Cp is specific heat at constant pressure.

- line 109, This sentence describes the spikes in surface heat flow in Figure 1. While these are easy to spot for case 1, case 2 does not feature them clearly nor does case 3, which seems to feature only one peak, but then does not seem to have the same equilibrium state before and after the peak (heat flow is quite different). So, are these heat flow peaks really characteristic for the discussed regime?

Response: This is a valid point. The peaks are characteristic of the strength of the upwelling flow – and so are weaker and/or more complicated where the upwelling flow is weaker and/or more complicated. As mentioned above, these simulations are meant to demonstrate the point of the paper, and that did not extend to actually modelling break-up processes of global supercontinents in detail. Therefore the details of the simulations are not critical; what they need to demonstrate is that a significant upflow will inevitably be associated with a significant near surface horizontal flow.

- line 152-155: This paragraph is extremely short and quite speculative. I don't think it is worth to call this a separate section. Since my other comments probably require additions to the manuscript, I actually suggest deleting this paragraph. If you want to keep this section, however, it should be extended and more explicitly linked to the discussed models.

Response: Yes – this paragraph is concise and is somewhat speculative but to our mind a very reasonable suggestion to make in the context of this manuscript. We prefer to retain the essence of the text but have reduced a little further and merged the text with original section 5.3, now 5.2.

Manuscript now reads: [Section 5] During break-up under the conceptual model presented here, margin segments located near active upwellings would show evidence of extensive magmatism, margin segments along-strike, where upwelling is not as concentrated, would be dominated by extension. Thus observations of both volcanic and non-volcanic margins during break-up could be satisfied (e.g. B. C. Storey 1995).

- line 167/168: How do you actually estimate from your results that the events take 10s of Myr? Just from the presented heat flow curves? Also, please add a reference here when you mention that this is comparable to Earth's timescales.

Response: The estimate is derived by using average model surface velocities, such as those shown in the new Figure 4 to scale the model time durations of the heat pulses in Figure 1, assuming an average Earth surface velocity of 5 cm/yr. This is necessarily approximate. We also refer to our earlier work.

Added to the manuscript: [Section 4] An estimate of the real Earth duration of these events was produced by taking the average (non pulse) surface velocities and deriving a scaling factor vs a real Earth velocity of 5 cm/yr, by which event durations could be evaluated. For example, Case 2 with $\sim$3 cm/yr velocity and the model event duration of $\sim$1x10^8 yr produces a real Earth duration of $\sim$60 Myr. We have used this approach previously to estimate durations between mantle avalanches (Wolstencroft, 2008).

- line 178: I suggest to add a reference to Yoshida's works here, too (e.g. Yoshida & Santosh., 2014, Geosci. Frontiers, 5, 77-81).

Response: Added.

- Table 1: Consider to include the symbols for the physical properties used in eq. (1) in the table, e.g. alpha for the thermal expansivity. I would also appreciate values for the reference viscosity, right now it is only given implicitly via Ra.

Response: We have added the reference viscosity values to Table 2. The symbols are already defined in the text and are in common usage.

- Figure 2: I think this figure would benefit from a time series of the "absolute radial mass flux" similar to the one of surface heat flow as already presented. Ideally, this should be done Figure 1, too, because then the reader could get an idea of how a mantle avalanche is linked to the surface heat flow spikes and perhaps get an idea about timescales, which would improve the (very short) discussion in section 5.3.

Response: We already present indications of radial mass flux through the models by depth at different snapshots (Fig. 2). We have decided not to include further quantitative analysis of absolute (global) radial mass flux vs time as it will distract from our core argument. Additional such figures are presented in Wolstencroft and Davies, (2011), and many similar related papers. Related to this, our addition of (new) Figure 4 provides surface velocities vs time, a much more useful measure in the context of our argument.

Some very minor suggestions: - line 8: "long-range" –> "long-wavelength" (?)

Response: long-range feels more intuitive to us in the context of an abstract.

- line 74: "The values" –> "The parameter values"

Response: amended

- line 93: "set with" –> "set up with" (?)

Response: altered to 'set by'

- line 101: "by the temperature change" –> "by the superadiabatic temperature change"

Response: amended

- line 107: delete "slightly"

Response: amended

- line 175: Check sentence structure: "including by diking"?

Response: section modified/simplified

Manuscript now reads: While the simple estimates above considered the driving force that an upwelling can provide, hot upwellings can also lead to magmatism, which can help to weaken the lithosphere e.g. by dyking (Bialas et al., 2010; Brune et al., 2013).

RC2

Main reviewer comment: The bottom line is that plate tectonics is an observed phenomenon but mantle avalanches are a phenomenon that appears in some mantle convection simulations and no concrete evidence for avalanches in the Earth's history exists. A study claiming that avalanches can play a role in supercontinent breakup should at least include model supercontinents and mantle convection observables (plates) as a starting point.

Response: Like the first reviewer, this comment while correct, misses the point of this manuscript, which we feel is clear in the text and abstract. The manuscript does not claim that avalanches have to play a role in supercontinental breakup, but rather uses them as an example mechanism that produces globally organised mantle flow, demonstrating the critical fact that a global process must be associated with a surface return flow – which would evidence itself as long-distance extension; and hence that passive and active mechanisms can be considered to act simultaneously for global supercontinent break-up. As in response to reviewer 1 we have made an effort to

bring this point through even more clearly, e.g. that is applies most strongly to global supercontinent breakup and will be much weaker for minor continental breakup.

Further comments on presentation: 1. The figures should include some model geotherms (i.e., laterally averaged temperature as a function of depth).

Response: If we wanted to present these models for their own right, then we agree there would be a benefit to present the model geotherm. In the context of the goal of this manuscript we do not see the value.

2. A figure of the surface viscosity field of model 3 should be included.

Response: Again, since (as mentioned in response to Reviewer 1) we are not explicitly modelling the plate break-up and that was not the goal of the variable viscosity model, we see no benefit to present the (near) surface viscosity field.

3. Present core heat flow as well as surface (Figure 1 is heat flow, not flux).

Response: If we wanted to present these models for their own right, then we agree there would be a benefit to present the core heat flow. In the context of the goal of this manuscript we do not see the value.

4. What is a 'smaller' convective feature (line 88) and why would it have more trouble breaking through the phase change? Specifically, it should be more vigorous in a high Rayleigh number flow.

Response: By smaller, we mean smaller in radius/planform. We alter the text slightly to make this clearer and point out that this effect is predicted by theory of Tackley and has been illustrated in a range of numerical simulations.

Manuscript now reads: High Ra produces shorter wavelength convective features, which by their weaker ability to counteract the negative buoyancy effect, are less able to break through the 660 phase change (Peltier, 1996; demonstrated by Tackley et al. 1993).

5. On line 67 a comparison appears to be made between 2D and spherical shells. Is that 2D Cartesian geometry or 2D annuli geometry (please be clear).

Response: The 2D models were Cartesian for Davies and Herein references, this has been updated in the text.

6. On line 118 what exactly is meant by inward and outward as they are used here. This doesn't appear to be referring to radial directions but rather lateral. Can more careful wording by offered?

Response: We have clarified by describing as "surface lateral" the inward flow and the outward flow as "radial" above the vertical upwelling.

Manuscript now reads: In terms of motion, this event produces globally organized surface-lateral flow towards the avalanche and radial flow above the antipodal upwelling. This motion is demonstrated in Supplementary Animation 1.

7. There are many previous studies on supercontinent breakup and identification of its causes that have not received adequate referencing. For example, please check and include some of the papers from the 90s, in particular those that discussed passive versus active mechanisms for supercontinent rifting.

Response: There are indeed many papers on Supercontinental breakup, but this is clearly not a review paper. We feel that we have sufficiently referenced earlier work – from the earliest work related to "passive" versus "active" continental breakup, through seminal papers, to some of the most recent contributions. If there are explicit suggestions then we would be happy to consider them (e.g. see further responses below). Note that we have added a number of additional references in response other suggested improvements.

SC1

1) An important aspect that should be mentioned is the mantle structure. In particular, should be mentioned that the temporal evolution of the convective wavelength is still an

enigmatic question, and it has important implications for the location of major mantle upwellings and the initiation of continental breakup. The present-day Earth's mantle structure is dominantly at spherical harmonic degree-2. However, how mantle structure may have evolved in the geological past is still unclear (Zhong et al., 2007). Also, phase transformations in the mid-mantle–which are often ignored in mantle convection simulations–introduce buoyancy forces and thermomechanical effects in the convective mantle system that can fundamentally affect the patterns of mantle convection (Faccenda & Dal Zilio, 2017).

Response: This is an interesting point, but a little away from the goal of the manuscript. Since it does though relate to the bigger questions raised by us, we have included the reference in the Discussion section, Zhong et al., (2007) has of course already been included.

Inserted in the Manuscript ∼ln 215: It is likely that the pattern of mantle convection has evolved over time and that other factors will influence the detail (e.g. Faccenda and Dal Zilio, 2017).

2) I am intrigued by the discussion about the source of these far-field extensional stresses. The authors should consider that subduction and subsequent roll-back of oceanic plates at continental margins have been invoked to support the occurrence of large-scale, long-term extensional stresses (Bercovici & Long, 2014; Lowman & Jarvis, 1996). Also, numerical simulations indicate that downgoing slabs play a central role in establishing the location and formation of subcontinental mantle plumes (Tan et al., 2002; Heron & Lowman, 2011; Heron et al., 2015; Lenardic et al., 2011; Zhang et al., 2010; Zhong et al., 2007).

Response: Again an interesting point, we have included some of the references mentioned in the Discussion section relating to fringing subduction ∼ln 210 - some were already included. We are of the view that the organisation of plumes by slabs is somewhat off topic for this manuscript but acknowledge that it is an interesting related ques-

tion.

3) During the last years, the role of deep slab dynamics appears central to triggering breakup of continents. Penetration of the slab into the lower mantle may generate a surge of compression at the plate boundary because (i) trench migration slows down (Goes et al., 2008; Faccenna et al., 2017), and (ii) because the upper plate is dragged against the subduction zone by a large-scale return flow. This return flow seems to be the key ingredient to triggering supercontinent breakup: subduction of lithosphere in the lower mantle reorganises the mantle flow into a wide cell, thereby localising extensional stresses at greater distances from the trench (Dal Zilio et al., 2017), which eventually may culminate in the breakup of supercontinents.

Response: This idea has very similar elements to the idea we present here and is very recent – 2 of the suggested papers were published simultaneously with our submission!, we are happy to be able to include a paragraph linking to this work.

Included in the Manuscript: [Section 5.3] Aside from mantle avalanches – used as an example mechanism here – subduction reorganisation offers another possible mechanism to produce global scale flow to cause supercontinent break-up. Goes et al. (2008) demonstrated how slabs could 'pile up' or 'lay down' above 660, causing subduction to be constrained within the upper mantle and to slow down, potentially leading to stresses sufficient to build mountain chains (Faccenna et al., 2017). When slabs do sink into the lower mantle, the length scale of subduction can increase. If sufficient material is involved, a lateral flow regime similar to that produced by mantle avalanche mechanism is set up – producing sufficient stress over durations that could lead to supercontinent break-up (Dal Zilio et al., 2017).
* * *
[Figure]

**Fig. 1.** New Figure 4